# The CBS/H₂S signalling pathway regulated by the carbon repressor CreA promotes cellulose utilization in *Ganoderma lucidum*
Jiaolei Shangguan, Jinjin Qiao, He Liu, Lei Zhu, Xiaofei Han, Liang Shi, Jing Zhu, Rui Liu ⓘ , Ang Ren & Mingwen Zhao ⓘ ✉

Cellulose is an important abundant renewable resource on Earth, and the microbial cellulose utilization mechanism has attracted extensive attention. Recently, some signalling molecules have been found to regulate cellulose utilization and the discovery of underlying signals has recently attracted extensive attention. In this paper, we found that the hydrogen sulfide ($H_2S$) concentration under cellulose culture condition increased to approximately 2.3-fold compared with that under glucose culture condition in *Ganoderma lucidum*. Further evidence shown that cellulase activities of *G. lucidum* were improved by 18.2-27.6% through increasing $H_2S$ concentration. Then, we observed that the carbon repressor CreA inhibited $H_2S$ biosynthesis in *G. lucidum* by binding to the promoter of *cbs*, a key gene for $H_2S$ biosynthesis, at "CTGGGG". In our study, we reported for the first time that $H_2S$ increased the cellulose utilization in *G. lucidum*, and analyzed the mechanism of $H_2S$ biosynthesis induced by cellulose. This study not only enriches the understanding of the microbial cellulose utilization mechanism but also provides a reference for the analysis of the physiological function of $H_2S$ signals.

Cellulose acts as the major component of plant cell walls as well as an abundant renewable carbohydrate[1,2]. Because resource is currently being exhausted, further effective utilization of cellulose is of great significance. Cellulose can be utilized by microorganisms as a low-cost and sustainable source of carbon[1,3]. Thus, microbial cellulose utilization is beneficial to the sustainable use of energy and carbon cycles in the biosphere, and deserves emphasis. The present work provided a fundamental understanding of microbial cellulose utilization components, including cellulases, regulating transcription factors and regulating signaling pathways. Enzymes with cellulose hydrolysis functions include three types, cellobiohydrolases (CBHs), endoglucanases (EGs) and β-glucosidases (Bgs)[4,5]. Some transcription factors have also been reported to play roles in cellulose utilization in filamentous fungi, such as the positive transcriptional activators, XlnR, ClrA and ClrB[6–9], and the repressor, CreA[3,10–13]. Cellulase activity can be regulated by basal levels of enzyme production and transcription factors, which are widely recognized. However, the role of many underlying regulators involved in cellulose utilization remains to be explored.

To date, some signal transduction pathways have been reported to participate in the regulation of cellulase activities under different stimuli conditions with a variety of modes, such as the G protein signaling pathway, AMP-activated protein kinase (AMPK) signaling pathway, mitogen-activated protein kinase (MAPK) signaling pathway, $Ca^{2+}$ signaling pathway and the cyclic AMP-dependent protein kinase A (cAMP-PKA) signaling pathway[11,14–19]. For example, members of MAPK signaling pathway were involved in cellulose utilization in *Trichoderma reesei*, such as the negatively regulatory, Tmk1 and Tmk2, and the positively regulatory, Tmk3[15,20]. The $Ca^{2+}$ signaling pathway was activated by cAMP to improve cellulase expression in *T. reesei*[14,21]. In *Aspergillus nidulans*, PKA indirectly phosphorylated at S319 of CreA under glucose culture condition, which resulted in inhibiting entry of CreA into the nucleus and reducing the transcriptional inhibition of cellulase[19]. Our previous study investigated the involvement of some signaling pathways in cellulose utilization in *G. lucidum*. For example, GlSwi6B, a member of MAPK signal transduction pathways, significantly increased the concentration of cytosolic $Ca^{2+}$, thereby promoting the activities of cellulase and xylanase in *G. lucidum*[17].

Key Laboratory of Agricultural Environmental Microbiology, Ministry of Agriculture and Rural Affairs; Department of Microbiology, College of Life Sciences, Nanjing Agricultural University, Nanjing 210095 Jiangsu, PR China. ✉e-mail: mwzhao@njau.edu.cn

Glsnf1, a member of AMP-activated protein kinase, improved cellulase activity by reducing the transcription level of the *creA* gene[11]. These works add new insights into our understanding of cellulose degradation and are of great significance for the effective utilization of cellulose as a renewable carbohydrate resource. Therefore, exploring novel signaling molecules that regulate cellulose utilization is important.

Hydrogen sulfide ($H_2S$) is now recognized as an endogenous signaling gasotransmitter in various species[22,23]. Cystathionine β-synthase (CBS), which catalyzes the condensation of homocysteine and cysteine to produce $H_2S$[24]. $H_2S$ biosynthesis can be induced by multiple stresses. In *G. lucidum*, CBS-synthesized $H_2S$ was induced by heat stress and inhibited the heat-induced secondary metabolism accumulation[22]. In wine-producing *Saccharomyces cerevisiae*, $H_2S$ biosynthesis was induced in the absence of assimilable nitrogen[25], which suggests that $H_2S$ could be induced by nutrient deficiency. Furthermore, $H_2S$ exhibited multiple physiological functions in response to various stresses. For example, $H_2S$ acted as an antioxidant and antiapoptotic signal in animals[26]. $H_2S$ also acted as an antioxidant signal molecule to improve the growth rate of plants under $Cu^{2+}$ and heat conditions[27,28]. Interestingly, cysteine supplementation reduced the furfural-induced accumulation of reactive oxygen species (ROS) and increased biomass during lignocellulosic utilization through increasing $H_2S$ concentration in *Zymomonas mobilis*[29]. These results implied a potential role of $H_2S$ in cellulose utilization, while direct evidence remains unavailable. Therefore, the mechanism of $H_2S$ biosynthesis and physiological function under cellulose culture conditions need to be explored.

Fungi are well-known organic-decomposing agents, especially mushrooms, which can use cellulose substrates[1]. *G. lucidum* is an important large basidiomycete with both medicinal value and economic value. Genome sequencing studies have shown that *G. lucidum* contains one of the largest sets of wood-breaking enzymes among basidiomycetes[30]. Therefore, *G. lucidum* is a good material for studying the regulatory mechanism of cellulose utilization. However, the role of $H_2S$ signaling pathways in cellulose utilization remains unclear. In this study, we found that the $H_2S$ concentration was increased under cellulose culture conditions. Increasing $H_2S$ concentration through pharmacological and genetic means enhanced cellulase activity. Furthermore, carbon repressor CreA inhibited the expression of *cbs*, a gene encoding the $H_2S$ synthetic enzyme, and the biosynthesis of $H_2S$ under cellulose culture conditions. Further research found that CreA binds to the *cbs* promoter at "CTGGGG". Our study explored a novel signaling molecule, $H_2S$, which promotes the cellulose utilization in *G. lucidum*, and analyzed the mechanism of cellulose-induced $H_2S$ biosynthesis. It was beneficial not only to the cultivation of *G. lucidum* but also to the utilization of the most abundant carbon resources in the biosphere.

## Results

### The intracellular concentration of H₂S was increased by cellulose

To explore whether $H_2S$ signals respond to changes in the carbon source, $H_2S$ concentration was measured in *Ganoderma lucidum* under different carbon sources (glucose or microcrystalline cellulose) culture conditions. As shown in Fig. 1a, b, we observed that the fluorescence of $H_2S$, measured by SF7-AM fluorescence probe, was significantly ($p < 0.01$) increased to ~2.3-folds under cellulose culture condition compared with that under glucose culture condition. This result shows that the intracellular $H_2S$ concentration of *G. lucidum* is increased by cellulose culture condition.

The gene transcription levels of 10 putative $H_2S$ biosynthetic enzymes were also measured. As shown in the heatmap, the expression levels of *lcd1*, *cse1* and *cbs* were significantly ($p < 0.001$) increased (Fig. 1c). Among them, the *cbs* expression level was significantly ($p < 0.001$) increased to ~13.3-fold under cellulose culture condition compared with that under glucose culture condition (Fig. 1c), which exhibits the most pronounced response to cellulose and implies that the cystathionine β-synthase (CBS) may be involved in regulating $H_2S$ biosynthesis under cellulose culture condition.

Then, a *cbs* gene overexpression vector was constructed (Supplementary Fig. 1a) and transfected into *G. lucidum*. Two *cbs*-overexpressed strains (*cbs-oe8* and *cbs-oe26*) were selected because the relative *cbs* mRNA content

increased ~3.8–4.4-fold (Supplementary Fig. 1b). The $H_2S$ concentration and the *cbs* expression level were detected in wild-type (*wt*), *cbs*-silenced, *cbs*-overexpressed and *sicontrol* strains under glucose or cellulose culture conditions. The fluorescence of $H_2S$ and the expression level of *cbs* gene in *cbs*-silenced strains significantly ($p < 0.05$) decreased by ~54.1–54.7% and 88.6–90.3% compared with that in the *wt* strain under cellulose culture condition, and significantly ($p < 0.05$) increased in *cbs*-overexpressed strains by ~70.5–76.3% and 148.7–164.6% (Fig. 1d, e, f), which indicated that CBS promoted $H_2S$ biosynthesis in *G. lucidum* under cellulose culture condition. In addition, cellulose culture condition result in a significant ($p < 0.05$) increase in the fluorescence of $H_2S$ in *cbs*-silenced strains compared with glucose culture conditions to ~1.3-folds, which was less than that in the *wt* strain, and no significant ($p > 0.05$) change in the expression level of *cbs* gene (Fig. 1d, e, f).

These results suggest that cellulose promotes the $H_2S$ biosynthesis in *G. lucidum*, and CBS might be one of the main $H_2S$ biosynthetic enzymes under cellulose culture condition.

### H₂S enhanced cellulase activity and cellulose utilization in *G. lucidum*

To explore the effect of $H_2S$ signal on cellulose utilization, cellulase activity was measured in the presence of sodium hydrosulfide (NaHS, a $H_2S$ donor) and hypotaurine (HT, a $H_2S$ scavenger), at a concentration that can significantly alter the intracellular $H_2S$ content in *G. lucidum* (Supplementary Fig. 2). As shown in Fig. 2a, the addition of NaHS increased cellulase activity, and the promoting effect increased gradually with increasing NaHS concentration. The addition of 60 μM NaHS significantly ($p < 0.001$) increased cellulase activity by ~27.6% compared with no treatment (Fig. 2a). The addition of HT inhibited cellulase activity, and the inhibitory effect was more evident with increasing HT concentration (Fig. 2b). HT (2 mM) significantly ($p < 0.001$) reduced cellulase activity by ~18.6% compared with no treatment (Fig. 2b). Pharmacological experiments suggest that $H_2S$ enhances cellulase activity in *G. lucidum* under cellulose culture condition.

To explore the potential influence of $H_2S$ biosynthesized by CBS on cellulose utilization, cellulase activity was measured in *wt*, *cbs*-silenced, *cbs*-overexpressed and *sicontrol* strains in the presence of NaHS and HT. Cellulase activity in *cbs*-silenced strains significantly ($p < 0.001$) reduced by ~31.9–32.2% compared with that in the *wt* strain (Fig. 2c). NaHS addition significantly ($p < 0.01$) increased cellulase activity of *cbs*-silenced strains by ~27.1–28.2% compared with under cellulose culture condition alone (Fig. 2c). Cellulase activity in *cbs*-overexpressed strains significantly ($p < 0.001$) increased by ~18.2–18.7% compared with that in the *wt* strain (Fig. 2d). HT addition significantly ($p < 0.001$) reduced cellulase activity in *cbs*-overexpressed strains by ~17.8-18.6% with under only cellulose culture condition (Fig. 2d). To demonstrate the potential influence of CBS on cellulose utilization of *G. lucidum*, wood chips were used as a main carbon source to cultivate *wt*, *cbs*-silenced, *cbs*-overexpressed and *sicontrol* strains for 15 days, and the growth length were measured. As shown in Fig. 2e, f, the growth length of *cbs*-silenced strains significantly ($p < 0.001$) decreased by ~53.4–56.1% compared with *wt* strain, while *cbs*-overexpressed significantly ($p < 0.01$) increased by ~14.4–16.4%. These results suggest that CBS-synthesized $H_2S$ improves the cellulase activity and cellulose utilization.

These combined results suggest that $H_2S$ improves the cellulase activity of *G. lucidum* under cellulose culture condition.

### Y1H screening for regulators of the *cbs* gene identifies diverse transcription factors

To further understand the mechanism of cellulose induced $H_2S$ biosynthesis, *cbs* promoter was analyzed. The intervening region between the *cbs* gene and upstream gene is very short (551 bp). Therefore, a total potential *cbs* promoter (+1 to −551 pb) was used as bait in a yeast one-hybrid (Y1H) library screen. The Y1H assay indicated that a total of 13 putative transcription factors (TFs) may directly bind to the *cbs* promoter (Supplementary Table 1). These putative TFs might regulate *cbs* gene transcription and $H_2S$ biosynthesis in *G. lucidum* under different conditions. Among

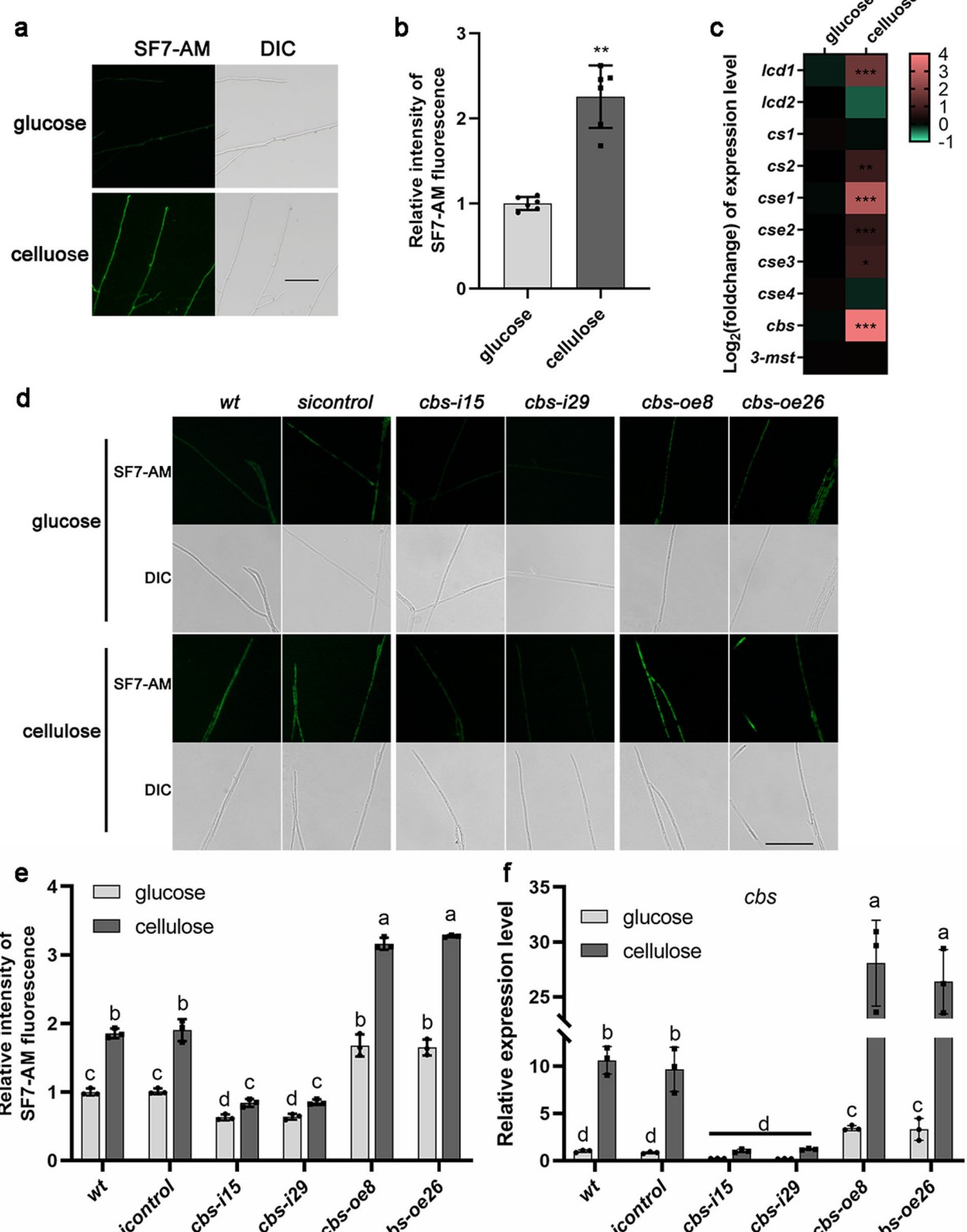

**Fig. 1 | Biosynthesis of H₂S was improved under cellulose culture condition.**
**a**, **b** Change in the H₂S concentration was measured by SF7-AM fluorescence probe in wild-type (*wt*) strain under glucose or cellulose culture condition. The average fluorescence intensity values of all mycelia in the 6 photos were quantified. Scale bar = 100 μm. **c** Log₂(foldchange) of genes expression level of putative H₂S biosynthetic enzymes: L-cysteine desulfhydrase (*lcd1* and *lcd2*), cysteine synthase (*cs1* and *cs2*), cystathionine γ-lyase (*cse1*, *cse2*, *cse3* and *cse4*), cystathionine β-synthase (*cbs*), 3-mercaptopyruvate sulfurtransferase (*3-mst*). The different letters indicate

significant differences between the lines ("*" means $p < 0.05$, "**" means $p < 0.01$, "***" means $p < 0.001$, according to Student's *t* test). **d**, **e** Change in the H₂S concentration was measured by SF7-AM fluorescence staining in *wt*, *sicontrol*, *cbs*-silenced and *cbs*-overexpressed strains under glucose and cellulose culture condition. Scale bar = 100 μm. **f** Relative expression level of *cbs* gene in *wt*, *sicontrol*, *cbs*-silenced and *cbs*-overexpressed strains, cultured under glucose and cellulose condition. The different letters indicate significant differences between the lines ($p < 0.05$, according to Duncan's multiple range test).

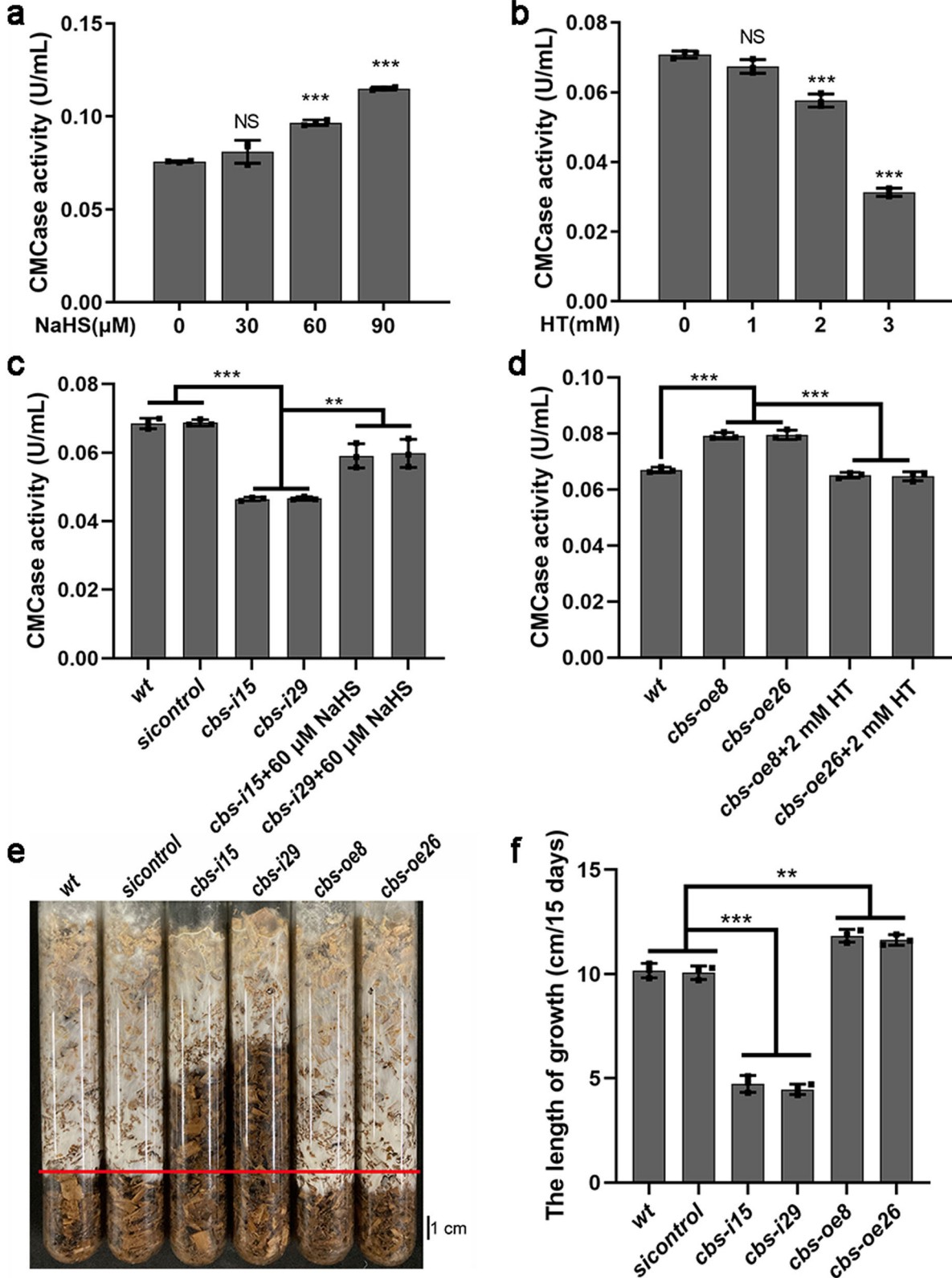

**Fig. 2 | H₂S enhances endocellulase activity. a, b** Endocellulase (CMCase) activities in *wt* strain under cellulose culture condition in the presence of sodium hydrosulfide (NaHS, H₂S donor) and hypotaurine (HT, H₂S scavenger). **c, d** Endocellulase (CMCase) activities in *wt*, *sicontrol*, *cbs*-silenced and *cbs*-overexpressed strains under cellulose culture condition in the in presence of NaHS and HT. **e, f** The growth length of *wt*, *sicontrol*, *cbs*-silenced and *cbs*-overexpressed strains after 15 days cultivated on wood chips. The red line marked the position of the *wt* strain grown on wood chips for 15 days. The different letters indicate significant differences between the lines ("**" means $p < 0.01$, "***" means $p < 0.001$, according to Student's t test).

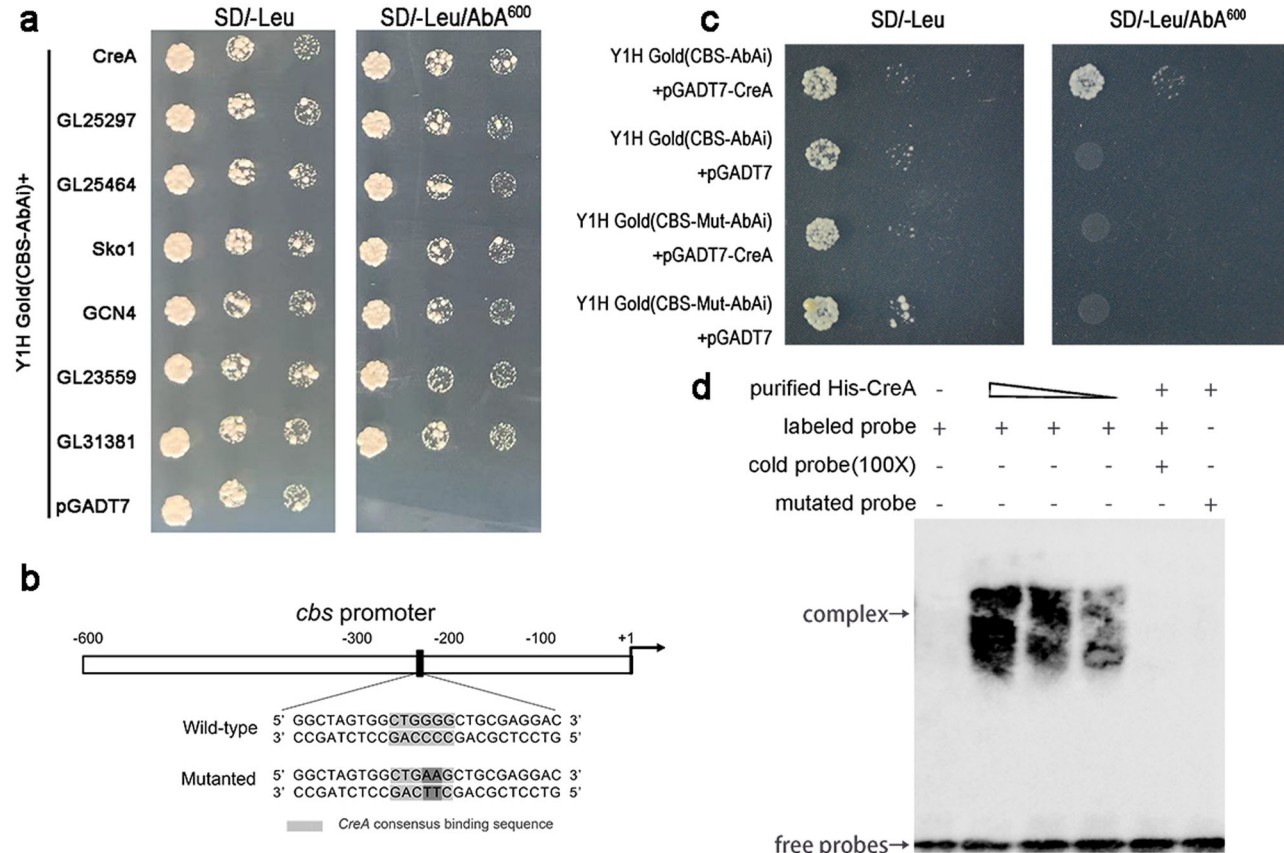

**Fig. 3 | CreA bound to *cbs* promoter at "CTGGGG". a** The result of highly conservative transcription factors (TFs) selected by a yeast one-hybrid (Y1H) library screen (n = 3). **b** Diagram of *cbs* promoter sequence. The black boxes indicate the binding sequence of CreA. **c** Y1H assay verification between CreA and *cbs* promoter. **d** Electrophoretic mobility shift assay (EMSA) for binding action of purified CreA or nuclear extraction with *cbs* promoter.

them, three TFs, had been reported in *G. lucidum*, were observed: CreA (mediated cellulose utilization of *G. lucidum*[11]), GCN4 and SKO1 (mediated nitrogen utilization of *G. lucidum*[31]), and four highly conserved TFs were observed: TFIIB, MCM1, Xbp1, and Crz1 (Fig. 3a and Supplementary Table 1). To explore the regulatory mechanism of *cbs* transcription and H$_2$S biosynthesis under cellulose culture condition in *G. lucidum*, CreA, a classical carbon catabolite repressor, was selected for further study.

## CreA could bind to the *cbs* promoter
To further determine the binding effect of CreA to the *cbs* promoter, Y1H assay and EMSA were performed. The Y1H assay indicated that compared with the negative control (pGADT7), Y1HGold yeast transformed pGADT7-CreA, developed obvious colonies (Fig. 3c). The EMSA result showed that compared with lanes without protein addition, an obvious binding complex band was observed in lanes with purified His-CreA protein addition (Fig. 3d). These binding effects were competitively inhibited by unlabeled cold probes (Fig. 3d). Gradually decreased binding complex bands were observed between 5' biotin-labeled probes of the *cbs* promoter and purified His-CreA protein in lanes with a decrease in protein concentration (100 mM, 50 mM, 20 mM) (Fig. 3d). These results suggest that CreA can bind to the *cbs* promoter.

To further explore the binding site of CreA in the *cbs* promoter, the *cbs* promoter was analyzed, and observed a typical CreA binding site, "CTGGGG", at -232 bp to -226 bp (Fig. 3b). Then, two GG nucleotides in the middle of the conserved binding domain were mutated to AA nucleotides as a previous study[32]. As shown in results of the Y1H assay, compared with the positive control (CBS-AbAi), Y1HGold yeast, transformed CBS-Mut-AbAi, did not develop significant colonies (Fig. 3c). The EMSA showed that compared with the positive control (non-mutated 5' biotin-labeled probe), the mutated 5' biotin-labeled probe could not bind with purified

CreA protein (Fig. 3d). These results indicate that CreA can bind to the *cbs* promoter at "CTGGGG".

## CreA inhibits *cbs* gene expression and reduces H$_2$S biosynthesis
To explore the regulatory role of CreA in the CBS/H$_2$S signaling pathway in *G. lucidum*, a *CreA*-overexpress vector was constructed (Supplementary Fig. 3a) and transfected into *G. lucidum*. Two *creA*-overexpressed strains (*creA-oe6* and *creA-oe8*) were selected for further analyses. The relative mRNA content level significantly ($p < 0.001$) increased about 5.7–6.0-fold and the relative protein content level significantly ($p < 0.001$) increased by ~52.7–54.5% in *creA*-overexpressed strains compared with the *wt* strain (Supplementary Fig. 3b, Supplementary Fig. 6, Fig. 4a, b). The relative mRNA and protein content levels in *creA*-silenced strains, previously established[22], significantly ($p < 0.001$) reduced by ~74.8–75.6% and 32.6–33.7% compared with those in the *wt* strain, respectively (Supplementary Fig. 3b, Supplementary Fig. 6, Fig. 4a, b).

Transcription levels of the *cbs* gene and fluorescence levels of H$_2$S were measured in *wt*, *creA*-silenced, *creA*-overexpressed and *sicontrol* strains. Compared with that in the *wt* strain, the transcription level of the *cbs* gene in *creA*-silenced strains significantly ($p < 0.001$) increased to ~2.1-folds and the fluorescence of H$_2$S was significantly ($p < 0.01$) increased by ~47.5–49.2% (Fig. 4c, d, e). The transcription level of the *cbs* gene and the fluorescence of H$_2$S in *creA*-overexpressed strains significantly ($p < 0.01$) decreased by ~44.3–48.3% and 29.1–30.3%, respectively (Fig. 4c, d, e). These results indicate that CreA reduces *cbs* gene transcription levels and intracellular H$_2$S biosynthesis.

## CreA inhibits *cbs* gene expression and reduces H$_2$S biosynthesis under cellulose culture condition
To explore the role of CreA in *cbs* expression and H$_2$S biosynthesis, the binding effect of CreA to the *cbs* promoter was measured under glucose

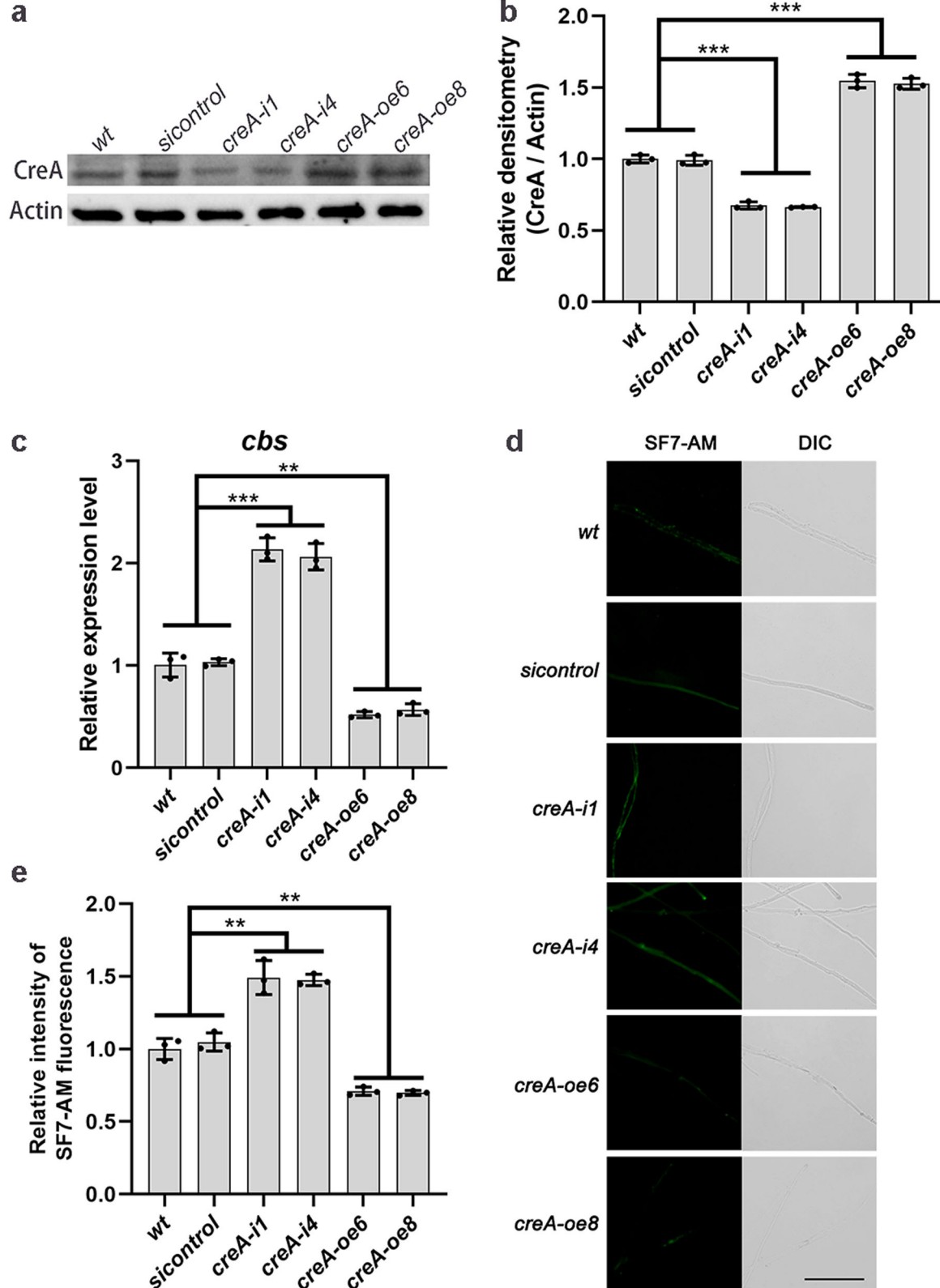

**Fig. 4 | CreA inhibits *cbs* transcription and reduces H$_2$S biosynthesis.**
**a**, **b** Immunoblot analysis of CreA proteins in *wt*, *sicontrol*, *creA*-silenced and *creA*-overexpressed strains, cultured on CYM medium for 7 days. **c** Relative expression level of *cbs* gene in *wt*, *sicontrol*, *creA*-silenced and *creA*-overexpressed strains, cultured on CYM medium for 7 days. **d**, **e** Change in the H$_2$S concentration was measured by SF7-AM fluorescence staining in *wt*, *sicontrol*, *creA*-silenced and *creA*-overexpressed strains, cultured on CYM medium for 7 days, measured by SF7-AM. Scale bar = 100 μm. The different letters indicate significant differences between the lines ("**" means $p < 0.01$, "***" means $p < 0.001$, according to Student's t test).

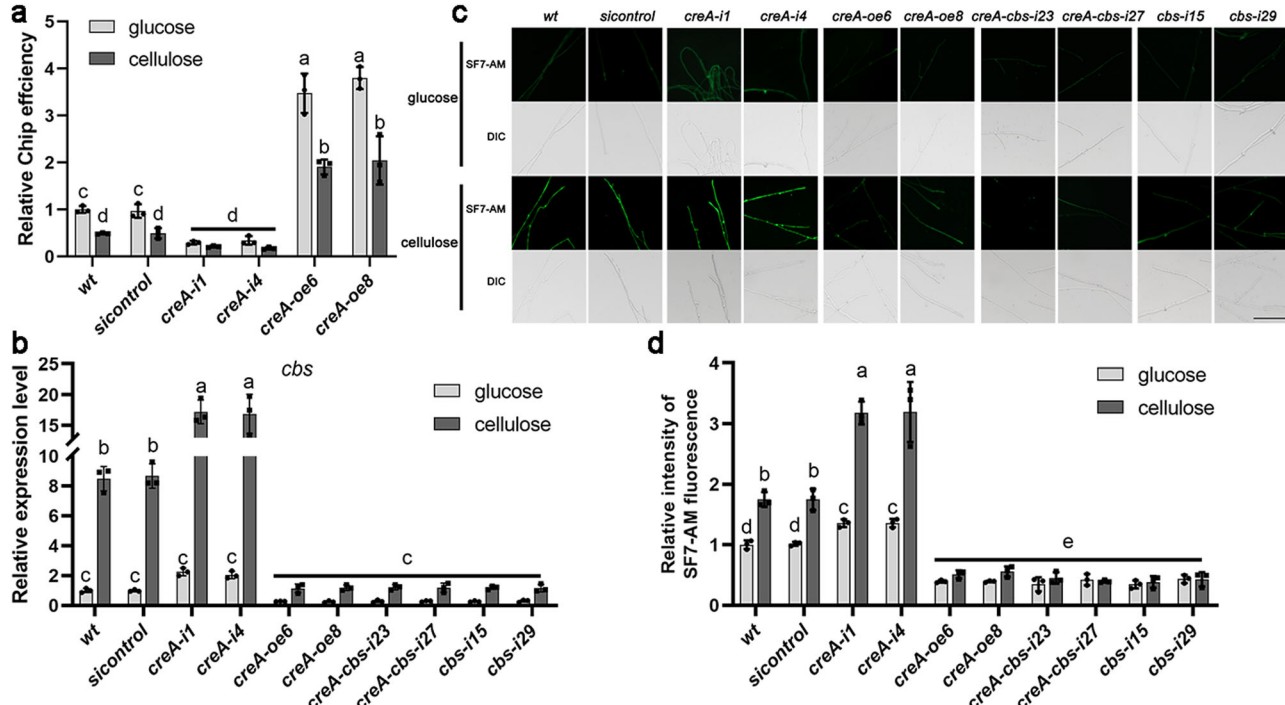

**Fig. 5 | Cellulose culture condition activates CBS/H₂S signal pathway by reducing CreA binding to the *cbs* promoter. a** ChIP assays for relative binding action with *cbs* promoter in *wt*, *sicontrol*, *creA*-silenced and *creA*-overexpressed strains, cultured under glucose and cellulose condition. **b** Relative expression level of *cbs* gene in *wt*, *sicontrol*, *creA*-silenced, *creA*-overexpressed and *creA-cbs*-silenced strains, cultured under glucose and cellulose condition. **c, d** Change in the H₂S concentration was measured by SF7-AM fluorescence staining in *wt*, *sicontrol*, *creA*-silenced, *creA*-overexpressed and *creA-cbs*-silenced strains, cultured under glucose and cellulose condition. Scale bar = 100 μm. The different letters indicate significant differences between the lines ($p < 0.05$, according to Duncan's multiple range test).

and microcrystalline cellulose culture condition. From the results of ChIP-qPCR, we found that cellulose culture condition significantly ($p < 0.05$) reduced the binding of CreA to the *cbs* promoter in the *wt* strain by ~51.1% compared with glucose culture condition (Supplementary Fig. 4 and Fig. 5a), indicating that cellulose culture conditions inhibited the DNA binding activity to the *cbs* promoter of CreA. In addition, under cellulose culture condition, the binding of CreA to the *cbs* promoter was significantly ($p < 0.05$) reduced by ~56.9–64.4% in *creA*-silenced strains and significantly ($p < 0.05$) increased to ~3.9–4.2-fold in *creA*-overexpressed strains compared with that in the *wt* strain (Fig. 5a). Then, transcription levels of the *cbs* gene and fluorescence levels of H₂S were measured in the *wt*, *creA*-silenced, *creA*-overexpressed and *sicontrol* strains. Under cellulose culture condition, the transcription level of the *cbs* gene in *creA*-silenced strains was significantly ($p < 0.05$) increased to ~2.0-folds compared with that in the *wt* strain, and the fluorescence of H₂S was significantly ($p < 0.05$) increased to ~1.8-folds (Fig. 5b, c, d). The transcription level of the *cbs* gene in *creA*-overexpressed strains under cellulose culture condition was significantly ($p < 0.05$) reduced by ~85.9–86.5% compared with that in the *wt* strain under, and the fluorescence of H₂S was significantly ($p < 0.05$) reduced by ~68.8–70.8% (Fig. 5b, c, d). These results indicate that cellulose inhibit the DNA binding activity to the *cbs* promoter of CreA, thereby enhancing *cbs* transcription and H₂S biosynthesis under cellulose culture conditions.

### CreA inhibits the positive regulation of CBS/H₂S in cellulose utilization of *G. lucidum*

To explore the role of H₂S and CreA in cellulose utilization of *G. lucidum*, a *creA-cbs*-silenced plasmid was constructed (Supplementary Fig. 5a). Two *creA-cbs*-silenced strains (*creA-cbs-i23* and *creA-cbs-i27*) were selected because of ~53.4–55.7% and 53.3–58.2% significant ($p < 0.01$) reduction in the relative mRNA content levels of the *creA* and *cbs* genes, respectively (Supplementary Fig. 5b, c). The transcription level of the *cbs* gene and the fluorescence of H₂S in *creA-cbs-i23* and *creA-cbs-i27* strains under cellulose culture condition significantly ($p < 0.01$) reduced by ~85.6–86.0% and

74.1–77.5% compared with those in *creA*-silenced strains, respectively, but no significant ($p > 0.05$) difference was observed compared with those in *cbs*-silenced strains (Fig. 5b, c, d).

Cellulase activity in the *wt*, *creA*-silenced, *creA*-overexpressed, *creA-cbs*-silenced and *sicontrol* strains was measured in the presence of NaHS and HT treatment. Cellulase activity in *creA*-silenced strains was significantly ($p < 0.05$) increased by ~54.6–58.1% compared with that in the *wt* strain, and significantly ($p < 0.05$) decreased by ~35.6–38.2% in *creA*-overexpressed strains (Fig. 6a). This result suggested that CreA inhibits cellulase activity. When *creA*-overexpressed strains treated with NaHS, the cellulase activity significantly ($p < 0.05$) increased by ~27.6–29.6% compared with that no treatment (Fig. 6a). Cellulase activity in *creA-cbs*-silenced strains were significantly ($p < 0.001$) reduced by ~59.7–61.7% compared with that in *creA*-silenced strains, but there was no significant difference compared with that in *cbs*-silenced strains (Fig. 6a). These results suggest that the silencing of both *creA* and *cbs* can inhibit the *creA* silencing-induced decrease in cellulase activity.

Then, the growth rates of the *wt*, *creA*-silenced, *creA*-overexpressed, *creA-cbs*-silenced and *sicontrol* strains on wood chips were further tested. As shown in the Fig. 6b, c, the growth rates of the *creA*-overexpressed and *creA-cbs*-silenced strains were significantly ($p < 0.05$) reduced by ~52.0–54.4% and 51.4–53.1% compared to that of the *wt* strain, while the growth rate of the *creA*-silenced strain was significantly ($p < 0.05$) increased by ~22.1–23.1%. These results suggest that CreA reduces cellulose utilization of *G. lucidum* by inhibiting CBS/H₂S signaling pathway.

### Discussion

Cellulose utilization contributes to the carbon cycle in the biosphere. Microbial cellulose utilization is widely used in both agriculture and industry[17,33]. Exploring the potential molecular regulatory mechanisms of cellulose utilization in *Ganoderma lucidum*, a medicinal and edible basidiomycete, is not only be crucial to understanding the mechanism of mushroom growth and development, but also provides a theoretical basis

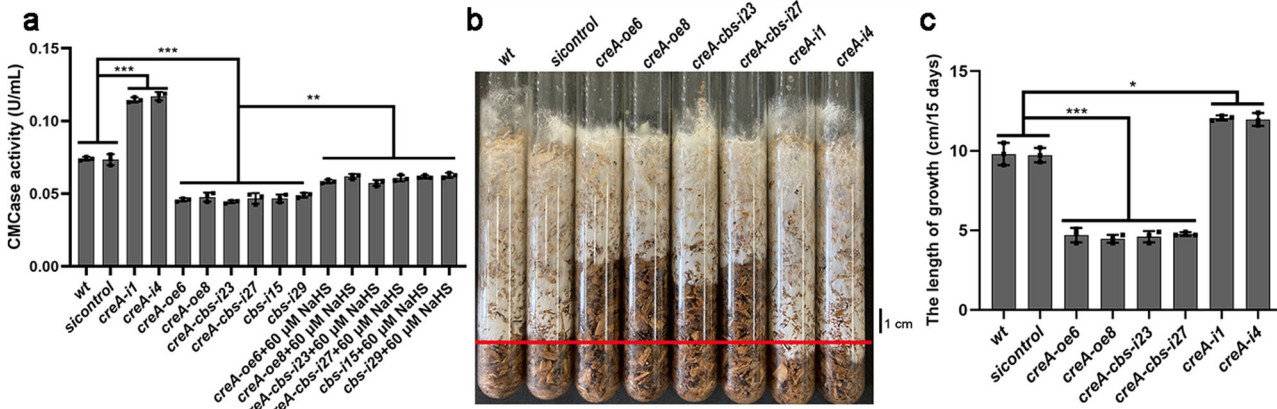

**Fig. 6 | CreA inhibits the positive regulation of CBS/H₂S in cellulose utilization of _G. lucidum_. a** CMCase activities in _wt_, _sicontrol_, _creA_-silenced, _creA_-overexpressed, _creA-cbs_-silenced and _cbs_-silenced strains under cellulose culture condition in the presence of NaHS. **b**, **c** The growth length of _wt_, _sicontrol_, _creA_-silenced, _creA_- overexpressed and _creA-cbs_-silenced strains after 15 days cultivated on wood chips. The red line marked the position of the _wt_ strain grown on wood chips for 15 days. The different letters indicate significant differences between the lines ("\*" means $p < 0.05$, "\*\*" means $p < 0.01$, "\*\*\*" means $p < 0.001$, according to Student's t test).

for other cellulose-utilizing microorganisms. The efficiency of cellulose utilization dependent on the regulation of transcription factors and the activity of cellulase is widely recognized. However, recent studies have found that some signaling pathways could also take part in the regulation of cellulose utilization, such as $Ca^{2+}$, cAMP and MAPK signals[14,15,17]. Therefore, the underlying molecular regulatory mechanism of cellulose utilization in _G. lucidum_ deserves emphasis and awaits further investigation. In this paper, we depict the promoting effect of hydrogen sulfide (H₂S) on the activity of cellulase in _G. lucidum_ for the first time. Furthermore, H₂S biosynthesis was negatively regulated by the carbon repressor transcription factor, CreA. Our results provided new insights into the mechanism of microbial cellulose utilization.

In this work, we observed that H₂S improved the cellulose utilization in _G. lucidum_. Before this work, the role of other signaling molecules on cellulase activity and its mechanism were also reported, such as cAMP and $Ca^{2+}$[14,17,21,34]. For instance, cAMP and $Ca^{2+}$ improved the expression of cellulase in _Trichoderma reesei_[14,21]. Increasing cytosolic $Ca^{2+}$ concentration promoted the activities of cellulase and xylanase in _G. lucidum_[17]. The silence of _PoLaeA2_ and _PoLaeA3_, key global regulators, leads to a reduction in cytosolic $Ca^{2+}$ content and cellulase activity in _Pleurotus ostreatus_[34]. In addition, H₂S has also been widely reported to have complex interactions with various signal molecules. For example, H₂S increases cAMP level in resting cells, while decreases cAMP level when adenylyl cyclases are activated[35]. Additionally, H₂S also acts as a regulator of the $Ca^{2+}$ channels to regulate the intracellular $Ca^{2+}$ concentration in various animal cells with different physiological effects[36–38]. H₂S enhance the tolerance of plants to drought stress by activating $Ca^{2+}$ signaling[39]. Our previous work demonstrated that H₂S leads to a decrease in cytosolic $Ca^{2+}$ level in _G. lucidum_ under heat stress[22]. Therefore, whether H₂S improve cellulase activity under cellulose culture conditions through interactions with other signaling pathways remains to be further studied.

In our previous work, we found that the silencing of _creA_ led to an increase in transcription levels of cellulase genes and cellulase activity in _G. lucidum_ under cellulose culture condition[11]. In this study, we observed that the cellulase activity in the _creA-cbs_ silenced strains were significantly decreased than that of the _wt_ strain (Fig. 6a). These results imply an important regulatory role of CBS-synthesized H₂S in cellulase activity. Previous studies have reported that H₂S affect protein function by a variety of potential mechanisms. Firstly, H₂S could directly regulate the activity of proteins by persulfidation. For example, H₂S regulates autophagy under endoplasmic reticulum stress in _Arabidopsis_ by persulfiding ATG18a, a core autophagy component[40]. Secondly, H₂S could regulate transcription factors by persulfidation, thereby affect the transcription

expression of multiple genes downstream of the transcription factor. For instance, H₂S involve in the regulation of cucurbitacin C synthesis in cucumber by increasing the persulfidation level of His-Csa5G156220 and His-Csa5G157230 (transcription factors) and transcriptionally activate Csa6G088690 (a key synthetase for CuC generation)[41]. Thirdly, H₂S also could regulate the upstream regulator of transcription factors by persulfidation, thereby indirectly affect the downstream transcription expression of multiple genes. For example, H₂S regulates ABA signaling by persulfidating SnRK2.6, promoting the activity of SnRK2.6 and its interaction with a transcription factor acting downstream of ABA signaling, ABF2, in guard cells[42]. In addition, these various mechanisms by which intracellular H₂S affects physiological processes often exist simultaneously. For instance, in plants, H₂S not only regulated transcription levels of drought-responsive genes but also induced the persulfidation level of proteins, which involved in cellular response to oxidative stress, hydrogen peroxide catabolism and so on, to involve in drought stress responses[43]. Therefore, further research is needed to clarify regulatory mechanisms of H₂S on cellulase activity in _G. lucidum_.

Intracellular H₂S biosynthesis was regulated by some transcription factors and induced by stress[44]. For example, transcription factor specificity protein 1 (Sp1) can bind to the promoter of both _cse_ and _cbs_ gene, two H₂S biosynthetic enzymes, to regulate H₂S biosynthesis in animals[45,46]. Nuclear factor (NF)-Y and upstream stimulatory factor 1 (USF-1) were also involved in the regulation of _cbs_ promoter activity in HepG2 cells[47]. The transcription factor OsNACL35 increased in H₂S concentration by directly upregulating the expression of _OsDCD1_ by binding to the promoter of _OsDCD1_ gene under salinity stress[48]. The regulation of transcription levels of H₂S biosynthetic enzymes had been reported in animals and plants, but less so in microorganisms. In this study, yeast one-hybrid (Y1H) library screening results revealed the underlying transcription factors of the _cbs_ gene, including CreA (mediated cellulose utilization of _G. lucidum_[11]), GCN4 and SKO1 (mediated nitrogen utilization of _G. lucidum_[31]), and four putative transcription factors (TFIIB, MCM1, Xbp1, Crz1). We further revealed that CreA combined with the _cbs_ promoter, transcriptionally provoked _cbs_ gene expression and improved H₂S biosynthesis under cellulose culture condition by pharmacological experiments and genetic experiments. The roles of other transcription factors need to be further studied. The potential binding of multiple transcription factors to the _cbs_ gene implied that H₂S may be induced by a variety of growing conditions in _G. lucidum_, and providing unlimited possibilities for the physiological function of H₂S signals under stress.

CreA, a carbon catabolite repressor, was previously reported to directly bind to the gene promoter of cellulolytic enzymes, inhibit the transcription

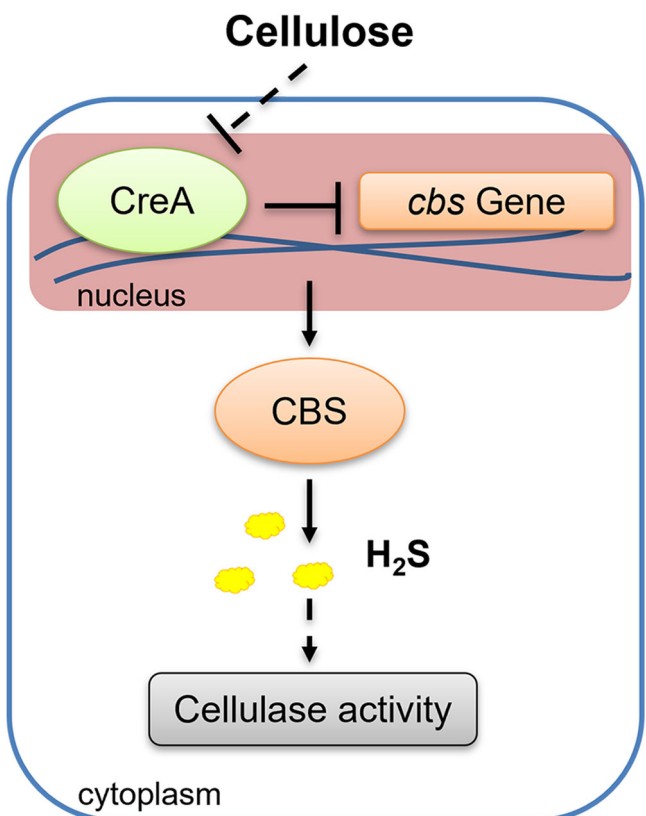

**Cellulose**

CreA — cbs Gene

nucleus

CBS

$H_2S$

Cellulase activity

cytoplasm

**Fig. 7 | Schematic representation.** The binding of CreA and *cbs* promoter reduced, and the transcription level of *cbs* gene increased, $H_2S$ biosynthesis promoted under cellulose culture condition. $H_2S$ improve the cellulase activity and cellulose utilization of *G. lucidum*.

of related genes, and regulate cellulose utilization in filamentous fungi, such as *T. reesei, Aspergillus nidulans, G. lucidum* and *Neurospora crassa*[3,11,12,49]. The transcriptomics analysis and the secretome analysis in *N. crassa* were observed that CreA repressed the expression level of genes, encoding enzymes involved in the utilization of alternative carbon sources, and cellulase activity under cellulose culture condition[50]. In *T. reesei*, CreA was considered as a regulator of the glucose assimilation rate[51]. Recent research has also shown that CreA could bind not only to the promoter of cellulose related genes, but also to the promoter of other genes to play a primary role in diverse physiological processes, such as carbon metabolism, secondary metabolism, iron homeostasis, oxidative stress response, development, N-glycan production, unfolded protein response, and nutrient and ion transport in *A. nidulans*[52]. Beyond carbon metabolism regulation, studies in various fungal species indicate additional CreA functions. For example, the ΔcreA mutants of *Aspergillus flavus*[53], *Magnaporthe oryzae*[54], and *Beauveria bassiana*[55] are affected in development and virulence, while repression of genes with functions in nitrogen uptake, development, chromatin remodeling, and the mediator complex depends on a functional CreA in *T. reesei*[51]. In *Aspergillus fumigatus*, the ΔcreA mutant impacts growth, fitness, and virulence[56]. In this study, we found that CreA negatively regulated *cbs* expression by binding to the *cbs* promoter, and inhibiting $H_2S$ biosynthesis, thereby reducing cellulase activity. Additionally, under repressing carbon sources culture condition, such as cellulose, the transcriptional level of CreA was decreased[32,49]. Therefore, CBS/$H_2S$ signaling pathway was provoked under cellulose culture condition in *G. lucidum*. $H_2S$ is a signaling molecule with multiple biological functions, and these results also implied that CreA may play a greater biological role through the regulation of $H_2S$ biosynthesis.

In this study, we revealed that $H_2S$ signals were promoted by activating *cbs* transcription level under cellulose culture condition (Fig. 7). Then, we observed that $H_2S$ improved cellulase activity in *G. lucidum* (Fig. 7).

Furthermore, we observed that CreA could bind to the promoter of *cbs* gene and reduce the transcriptional level of *cbs* gene (Fig. 7). In the case of repressing carbon sources, such as cellulose culture condition, CreA mRNA levels were decreased[32,49]. Therefore, cellulose reduced the inhibitory effect of CreA on the transcription level of *cbs* gene, thereby activating CBS/$H_2S$ signaling pathway under cellulose culture condition. These results explored a novel signaling molecule, $H_2S$, which participate in the regulation of cellulose utilization. Our study not only provides insight into the response mechanism of $H_2S$ signals in exposure to cellulose in *G. lucidum* but also benefits to the utilization of the most abundant energy resources in the biosphere.

## Methods
### Strains and culture conditions
The wild-type (*wt*) *Ganoderma lucidum* strain (obtained from Shanghai Academy of Agricultural Science) was used in a previous work[22]. The *G. lucidum* strains used in this experiment: *sicontrol*, *cbs*-silenced and *cbs*-overexpressed strains were established previously[22]; *creA*-silenced strains were established previously[11]; *creA*-overexpressed strains were established as described in Supplementary Fig. 1, *creA-cbs*-silenced strains were established as described in Supplementary Fig. 3.

The culture conditions of *G. lucidum* was used as previous works[22,57]. For wood chips culture, *G. lucidum* were cultured on edible mushroom cultivation material (60% wood chip, 20% cottonseed hull, 18% wheat bran, 1% sucrose and 1% gypsum) at 28 °C and for 15 days. For other detection, *G. lucidum* were cultured in glucose-containing nutrient-rich liquid CYM medium at 28°C and 150 rpm for 5 days, and then collected and changed into single carbon source (1% glucose or micro-crystalline cellulose) liquid MCM medium (0.46% $KH_2PO_4$, 0.05% $MgSO_4$-$7H_2O$, 0.5% $(NH_4)_2SO_4$, 2 ml/l trace elements) at 28°C and 150 rpm for 2 days.

### Transcription level detection
The transcription level of *cbs* and *creA* were detected by RT-qPCR according to method described previously[22]. Total RNAiso Plus (TaKaRa, Dalian, China) was used to extract total RNA and cDNA was reverse transcribed using a PrimeScript RT reagent kit (TaKaRa, Dalian, China). 18S rRNA was used as a reference to analyze the transcription levels of *cbs* and *creA*. The oligonucleotide primers used are listed in the Supplementary Table 2. The relative transcription levels of genes were determined using the $2^{-\Delta\Delta CT}$ method.

### Endogenous $H_2S$ concentration detection
$H_2S$ concentration in *G. lucidum* was detected after incubation with sulfidefluor-7 acetoxymethyl ester (SF7-AM) as described in a previous study[22,58]. The average fluorescence intensity values of all mycelia in each photo were quantified. Zeiss Axio Imager A1 fluorescence microscope was used to fluorescence image for all samples under the same microscopy settings. The average fluorescence intensity of mycelium was analyzed using ZEN software.

### Endocellulase activity (CMCase) detection
Endocellulase activity was detected according to previously described methods[11]. The culture supernatants were collected for endocellulase activity (CMCase) assays. In brief, citric acid buffer (50 mM, pH 4.8) and 2% (W/V) sodium carboxymethyl cellulose (CMC-Na) were added and the reaction was carried out at 50 °C for 30 min, and then detected optical density at 540 nm.

### Yeast one-hybrid assay
A yeast one-hybrid (Y1H) assay was performed using the matchmaker gold yeast one-hybrid library screening system (Clontech, China). The target sequence of *cbs* promoter (+1 to −551pb) was cloned and inserted into the pAbAi vector. The CBS-AbAi vector was digested with BstBI enzyme (TAKARA, China) and then transformed into yeast stains. Strains

**Article**

were grown on SD/-Ura media for 2 days and positive transformants were selected by PCR. Next, the minimal inhibitory concentration of aureobasidin A (AbA) was confirmed as follows. Y1HGold (CBS-AbAi) were suspended in 0.9% NaCl (OD600: 0.002) and dotted onto SD/-Ura medium with AbA (0, 100, 150, 200, 300, 400, 500, 600, 800, and 1000 ng ml$^{-1}$) for 3 days. AbA (600 ng ml$^{-1}$) completely suppressed the growth of yeast strains and was used for subsequent experiments.

The Y1H screen assay was performed as previously described[59]. The cDNA library (pGADT7 vector) was conducted by SMART cDNA production technology (oebiotech, China). Then, screen the cDNA library by cotransformation and transformed yeast stains are plated on SD/-Leu/+AbA to select for colonies.

For the transcriptional activation test, the full-length complementary cDNA of *creA* was cloned and inserted into the pGADT7 vector. The constructed plasmids (pGADT7-CreA) were then transformed into Y1HGold (CBS-AbAi) on SD/-Leu/+AbA medium for 3 days, and the positive transformants and were chosen and dotted as described above.

### Expression and purification of CreA
CreA protein was expressed and purified as previously described[11]. The *creA*-28a plasmid was transformed into *E. coli* BL21 (DE3), and protein expression was induced by 1 mM isopropyl β-D-thiogalactopyranoside (IPTG) at 28 °C for 4 h. Protein verification was performed using SDS-PAGE.

### Electrophoretic mobility shift assay
An electrophoretic mobility shift assay (EMSA) was performed as previously described[46,59]. Both forward (5′-TGGCTGGGGC-3′) and reverse (5′-GCCCCAGCCA-3′) primers of *cbs* promoter were biotin labeled at the 5' end. The mutated primers were 5′-TGGCTGTTGC-3′ (forward) and 5′-GCAACAGCCA-3′ (reverse) biotin labeled at the 5' end. For EMSA analysis, the biotin-labeled primers were incubated with purified CreA protein (10 μg, 5 μg, 2 μg, respectively) in 5 μl binding buffer [10 mM Tris-HCl (pH 8.0), 1 mM DTT, 0.1 mM EDTA, 50 mM KCl and 5% glycerol]. In addition, the mixture without protein was used as a negative control.

### Chromatin immunoprecipitation assay
Chromatin immunoprecipitation assay (ChIP) was performed as previously described[46]. In brief, cross-linking of DNA with proteins was performed with 1% formaldehyde. The nuclei were isolated, and sheared chromatin was prepared by sonication. The sheared chromatin was immunoprecipitated with rabbit polyclonal anti-CreA antibodies and rabbit serum (D601019, Sangon, China, negative control). Another aliquot of sheared chromatin without incubation with antibodies was prepared as input. As shown in Sigure S2, DNA, isolated from immunoprecipitation, was detected by PCR with primers for the *cbs* promoter region were 5′-TTGACGCG-GACGGACAT-3′ (forward) and 5′-GGGAAGATGGTGGCAGAA-3′ (reverse). The relative binding efficiency of CreA to the *cbs* promoter was detected by RT-qPCR.

### Western blotting
Western blotting was performed as previously described[11]. Briefly, proteins from mycelia samples were separated in a 12% (w/v) SDS-PAGE gel and transferred to polyvinylidene difluoride membranes (Bio-Rad). The primary antibodies used to detect specific proteins in this report were anti-CreA (1:2000, rabbit polyclonal), anti-Actin (1:2000, mouse; CMCTAG) and anti-Histone3 (1:2000, AT0005, CMCTAG). ImageJ software was used to quantify the densities of the bands.

### Statistical analysis
The statistical analyses were performed using GraphPad Prism 8.0.2 (GraphPad Software, San Diego, CA, USA). All experimental data shown in this article were carried out in three independent samples to ensure that trends and relationships observed in cultures were reproducible. The error bars indicate the standard deviation from the mean of triplicates. The data were analyzed using Student's t test or Duncan's multiple range test. The $p < 0.05$ was considered significant.

### Reporting summary
Further information on research design is available in the Nature Portfolio Reporting Summary linked to this article.

## Data availability
All data generated or analyzed during this study are included in this published article (and its Supplementary Information files) or are available from the corresponding author on reasonable request. The source data underlying the graphs in the figure are shown in Supplementary Data 1. Uncropped western blots are in Supplementary Fig. 6.

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

## Acknowledgements

This work was supported by: China Agriculture Research System (project number CARS20), the National Natural Science Foundation of China (project numbers 31972059 and 32272787), and the Postgraduate Research and Practice Innovation Program of Jiangsu Province (project number KYCX220709).

## Author contributions

J.S. and M.W.Z. designed the study. J.S., J.Q., H.L., L.Z., and X.H. carried out experiments and analyzed data. L.S., J.Z., R.L., A.R., and M.W.Z. provided supervisor oversight. J.S. wrote the manuscript. All authors gave input and approved the manuscript.

## Competing interests

The authors declare no competing interest.
