## [Peer Review File · Communications Biology]

Reviewers' comments:

Reviewer #1 (Remarks to the Author):

The authors of this manuscript show that cellulose culture condition induces hydrogen sulfide (H₂S) synthesis in *Ganoderma lucidum*, and carbon catabolite repressor CreA inhibits this H₂S synthesis through repression of cystathionine β -synthase (CBS) gene expression. The authors also show that cellulase activity is increased by H₂S. These are interesting phenomena. However, the mechanisms of how cbs gene expression is induced by cellulose and how H₂S affects cellulase activity are not clear. These points are required for the readers of Communications Biology. In addition, there are various issues as indicated below.

Fig. 1b

I don't understand the y-axis label; what does 100% indicate?

Fig. 1b

The authors describe that H₂S concentration under cellulose culture condition increased by approximately 1.25-fold compared with that under glucose culture condition (lines 29-32 and 106-108). This description does not appear to be consistent with the results in Fig. 1b.

Fig. 1b

How many mycelia were examined?

Fig. 1d and 1e

The expression level of cbs gene should be examined.

Fig. 2a and 2b

The concentration of H₂S in the presence of NaHS or HS should be examined.

Fig. 3a

Not enough information is provided about Y1H screening. If the cDNA library was used for screening, information about the cDNA library and the cDNA fragments that were introduced into the positive clones should be provided. Or if the cDNAs of the transcription factors are cloned for screening, information about the transcription factor genes being cloned should be indicated.

Fig. 3b

I do not understand the significance of this phylogenetic tree. Is there any sense in combining different transcription factors into one phylogenetic tree?

Line 351 "positive"

Is this right?

The manuscript should be written more carefully because there are many minor errors.

Reviewer #2 (Remarks to the Author):

In this study, Jiaolei Shangguan and colleagues set out to investigate the intracellular signaling mechanisms involved in the utilization of cellulose by the fungus *Ganoderma lucidum*. First of all, cellulose is a highly abundant carbohydrate and understanding the mechanisms that take place during its degradation or conversion have direct importance to understand ecological aspects as well as a direct link to the biotechnological utilization of cellulose for biobased materials such as biofuels. On the other hand, *Ganoderma lucidum* is a mushroom-forming fungus with an important role in research and

folk medicine. Altogether, the topics under study in this manuscript are very relevant.

More specifically, the authors have demonstrated a link between cellulose utilization and hydrogen sulfide (H₂S) production by fungi. This link involved the direct binding of the transcription factor CreA, known to be important for carbon catabolite repression in fungi, to the promoter of the *cbs* gene, repressing its expression. The binding site was determined too.

The production of H₂S was increased on cellulose vs. glucose, but this increase was quite modest (1.25 folds), albeit significant. Exogenously added H₂S did enhance cellulase activity, so the link seems to be solid. Furthermore, the authors conducted some validation experiments with chemicals that activate and inhibit H₂S production as well as with silenced vs. overexpression cells. All data seem to fall into place and the rationale of the work is easy to understand. Nevertheless, some of the data is somewhat circumstantial and not necessarily indicative of a direct mechanism.

Fig.1: It would be important to present more controls for the SF7-AM method to measure intracellular accumulation of H₂S. Methods whose quantification is based on microscopy analysis strongly depend on appropriate controls so that autofluorescence is properly normalized. The same microscopy settings (e.g., exposure, gain, etc.) must be used for all samples to avoid situations of saturation, for instance. A non-staining control is missing, for instance. These controls could be a supplemental figure. Still related to this point, perhaps a more quantitative/robust method could have been tried, such as flow cytometry (in spores) or fluorimetry, using the SF7-AM probe.

Fig. 2c: In the *cbs*-silenced lines, either the gene silencing efficiency is modest or how can the authors integrate their conclusion that *cbs* is so important for cellulose degradation but at the same time *cbs*-silenced cells are able to increase CMCase activity when treated with NaHS? Perhaps justified because there are other important pathways playing a role? I think this should be discussed.

Fig. 2d: differences are quite modest, despite the statistical significance.

Fig. 3b: the phylogenetic is not showing informative data. Many more protein accessions would have to be used to make a more detailed study of the relationship between these transcription factors. Furthermore, the description of this figure in the legend is "evolutionary tree", which is an incorrect oversimplification. Additionally, the values next to the tree branches are not defined in the legend.

Minor changes

- Line 41: "maximum amount of renewable carbohydrates"; this phrase is unclear. What is meant by "maximum"?
- Line 76: "H₂S improved the tolerance of plants to Cu²⁺ and heat"; not clear or not specific enough. How did H₂S improve tolerance?
- Line 82: the word "well" seems unnecessary.
- Line 95: "a gene of H₂S synthesis enzyme"; this phrase is not properly written. Perhaps "a gene encoding the H₂S synthesis enzyme"? But could the authors be more specific about the role of this enzyme in the synthesis of H₂S?
- Line 97: "which promoting" is incorrect. "which promotes"?
- Fig. 2E: the meaning of the red, horizontal line should be specified in the figure legend.
- Line 287-305 seems more appropriate for the Introduction rather than Discussion

The manuscript is well written and easy to understand. However, an overall revision could be performed to fix minor incorrections.

Pedro Gonçalves

Reviewer #3 (Remarks to the Author):

General Comments:

The manuscript presents a well-executed study investigating the role of the hydrogen sulfide (H₂S) signaling pathway in *Ganoderma lucidum*, particularly its impact on cellulase activity and cellulose utilization. The research is meticulously conducted, and the findings contribute valuable insights into the molecular mechanisms underlying microbial cellulose utilization. The presented data is well-supported and convincingly demonstrates the involvement of the H₂S signaling pathway, specifically regulated by the carbon repressor transcription factor CreA.

Specific Comments:

Clarity and Structure:

The manuscript is well-structured, and the clarity of the presentation is commendable. The introduction effectively contextualizes the study, and the objectives are clearly defined. The results and discussion sections are logically organized, facilitating a smooth understanding of the experimental outcomes.

Experimental Design and Methods:

The experimental design is robust, and the methods are meticulously detailed, allowing for reproducibility. The use of both pharmacological and genetic experiments to probe the H₂S signaling pathway, as well as the involvement of CreA, enhances the study's credibility.

Data Presentation:

The data presentation is clear and concise. Figures and tables are appropriately utilized to illustrate key findings, and the inclusion of statistical analyses adds rigor to the interpretation of results. The use of fluorescence probes and gene expression analysis provides a comprehensive view of H₂S concentration and its regulatory mechanisms.

Discussion of Results:

The discussion of results is comprehensive, and the authors effectively relate their findings to existing literature. The identification of CreA as a negative regulator of H₂S biosynthesis adds novelty to the study. Additionally, the discussion on the potential influence of CBS on cellulose utilization is insightful.

Language and Style:

The language used in the manuscript is generally clear and concise. However, attention to minor grammatical issues, as identified in the detailed suggestions, will enhance the overall polish of the manuscript.

Conclusion:

The study significantly contributes to our understanding of the molecular mechanisms governing cellulose utilization in *G. lucidum*. The data is robust, the methodology is sound, and the results are well-discussed. With minor revisions for language and clarity, the manuscript is well-suited for publication in a peer-reviewed journal.

Here are some minor issues that needs to be addressed.

Some references regarding CreA homologs in other organisms should be included for example PMID: 21980519, PMID: 21619626, PMID: 31040248.

Here are some minor points with language use.

In line 104, "*Ganoderma lucidum*" could be italicized or written in a different format for emphasis on the species name.

In line 108, it should be "This result shows that" instead of "This result shown that."

In line 111, "genes transcription levels" should be "gene transcription levels."

In line 116, "cystathionine β -synthase (CBS)" could be introduced with "the" for better clarity, like "the cystathionine β -synthase (CBS)."

In line 118, "cbs-silenced and cbs-overexpressed strains" could be rewritten as "cbs-silenced, cbs-overexpressed, and control strains" for better clarity.

In line 123, "H₂S biosynthesis in *G. lucidum* and CBS might be one" might read more smoothly as "H₂S biosynthesis in *G. lucidum*, and CBS might be one."

In line 126, "To explore" could be followed by a phrase indicating the action, such as "To explore the effect of H₂S signal on cellulose utilization."

In line 136, "These results of pharmacological experiments suggested that" could be more concise, like "Pharmacological experiments suggested that."

In line 158, there's a space before the colon in "regulators of the cbs gene identifies diverse transcription." It should be "regulators of the cbs gene identifies diverse transcription:"

In line 218, "which indicated that cellulose culture condition" could be revised for clarity, like "indicating that cellulose culture conditions."

In line 234, "thereby improving cbs transcription and H₂S biosynthesis under cellulose culture condition" could be simplified, such as "thereby enhancing cbs transcription and H₂S biosynthesis under cellulose culture conditions."

In line 270, "Microbial cellulose utilization contributes to the carbon cycle in the biosphere and is widely used in both agriculture and industry" could be split into two sentences for clarity.

Dear Reviewers:

Thank you for your comments concerning our manuscript entitled “**CBS/H₂S signalling pathway regulated by the carbon repressor CreA promotes cellulose utilization in *Ganoderma lucidum*” (COMMSBIO-23-3923). Those comments are valuable and very helpful for revising and improving our paper and providing important guidance for our research. We have read through the comments carefully and revised our manuscript according to your and three reviewers' suggestions. We hope that the revised manuscript will be appropriate for publication.**

The revised sections are highlighted in red color in the revised manuscript.

Our point-by-point replies to the reviewers' comments are presented below.

Thank you very much for your attention and consideration.

Best regards.

Yours sincerely,

Mingwen Zhao

E-mail: mwzhao@njau.edu.cn

< COMMENTS FOR THE AUTHOR >

Reviewer #1:

The authors of this manuscript show that cellulose culture condition induces hydrogen sulfide (H₂S) synthesis in *Ganoderma lucidum*, and carbon catabolite repressor CreA inhibits this H₂S synthesis through repression of cystathionine β-synthase (CBS) gene expression. The authors also show that cellulase activity is increased by H₂S. These are interesting phenomena. However, the mechanisms of how *cbs* gene expression is induced by cellulose and how H₂S affects cellulase activity are not clear. These points are required for the readers of Communications Biology. In addition, there are various issues as indicated below.

Response: Thank you for your constructive comments during the review of the manuscript. Those comments are valuable and very helpful for our paper.

In this paper, we observed that the transcription of *cbs* genes is repressed by the transcription factor CreA. CreA was a recognized carbon catabolite repressor in filamentous fungi, such as *Trichoderma reesei*, *Aspergillus nidulans*, *Ganoderma lucidum* and *Neurospora crassa*¹⁻⁵. In the case of repressing carbon sources, such as cellulose culture condition, CreA mRNA levels were decreased^{3,6}. Therefore, we concluded that cellulose activities CBS/H₂S signalling pathway by reducing the binding of the repressor CreA to the *cbs* promoter under cellulose culture condition. These data and conclusions explain the mechanism by which cellulose induces the transcription of *cbs* genes. We also have added the related points in the Discussion (Page 18, Line 369-374).

As to how H₂S affects cellulase activity, we have only found this phenomenon. We have made some speculations in the Discussion (Page 15-16, Line 304-320). We also consider it is very important and necessary to assay the relevant mechanism. We will focus on the point in the future work.

Reference:

¹ Hu, Y. et al. In *Ganoderma lucidum*, Glsnf1 regulates cellulose degradation by inhibiting GICreA during the utilization of cellulose. *Environmental microbiology*, 22, 107-121, doi:10.1111/1462-2920.14826 (2020).

² Lichius, A., Seidl-Seiboth, V., Seiboth, B. & Kubicek, C. P. Nucleo-cytoplasmic shuttling dynamics of the transcriptional regulators XYR1 and CRE1 under conditions of cellulase and xylanase gene expression in *Trichoderma reesei*. *Molecular microbiology*, 94, 1162-1178, doi:10.1111/mmi.12824 (2014).

³ Ries, L. N., Beattie, S. R., Espeso, E. A., Cramer, R. A. & Goldman, G. H. Diverse regulation of the CreA carbon catabolite repressor in *Aspergillus nidulans*. *Genetics*, 203, 335-352, doi:10.1534/genetics.116.187872 (2016).

⁴ Lin, H., Hildebrand, A., Kasuga, T. & Fan, Z. Engineering *Neurospora crassa* for cellobionate production directly from cellulose without any enzyme addition. *Enzyme and microbial technology*, 99, 25-31, doi:10.1016/j.enzmictec.2016.12.009 (2017).

⁵ Sun, J. & Glass, N. L. Identification of the CRE-1 cellulolytic regulon in *Neurospora crassa*. *PloS one* 6, e25654, doi:10.1371/journal.pone.0025654 (2011).

⁶ Strauss, J. et al. The function of CreA, the carbon catabolite repressor of *Aspergillus nidulans*, is regulated at the transcriptional and post-transcriptional level. *Molecular microbiology*, 32, 169-178, doi:10.1046/j.1365-2958.1999.01341.x (1999).

Fig. 1b

I don't understand the y-axis label; what does 100% indicate?

Response: Thank you for your questions. We defined the fluorescence intensity value of Sulfidefluor-7 acetoxymethyl ester (SF7-AM) detected under glucose culture condition as 100%. Relative quantification was used to calculate the fluorescence values of SF7-AM detected under cellulose culture conditions. Corresponding description has been added in the Figure legends (Page 23, Line 474-475).

Fig. 1b

The authors describe that H₂S concentration under cellulose culture condition increased by approximately 1.25-fold compared with that under glucose culture condition (lines 29-32 and 106-108). This description does not appear to be consistent with the results in Fig. 1b.

Response: Sorry for any confusion caused by our unclear description. We have revised our statement in Page 2, Line 28-29 and Page 6, Line 114.

Fig. 1b

How many mycelia were examined?

Response: Thank you very much for your suggestions. We have added the description in the Methods (Page 19, Line 403-404) and the Figure legends (Page 23, Line 473-474). The average fluorescence intensity values of all mycelia in the 6 photos were quantified.

Fig. 1d and 1e

The expression level of *cbs* gene should be examined.

Response: Thank you for your suggestion. According to the suggestions, we have added the *cbs* gene expression levels of wild-type (*wt*), *si*control, *cbs*-silenced and *cbs*-overexpressed strains under cellulose or glucose culture conditions (the new Fig. 1f) and the description have been added in the revised manuscript (Page 7, Line 128-139) and the Figure legends (Page 23, Line 484-488).

Fig. 2a and 2b

The concentration of H₂S in the presence of NaHS or HS should be examined.

Response: Thank you for your attention and advice. We assume you're advising us to add the H₂S concentration in the presence of NaHS or HT. Therefore, we had added this data in the new Supplementary fig. 2 and the description have been added in Page 7, Line 146-147.

Fig. 3a

Not enough information is provided about Y1H screening. If the cDNA library was used for screening, information about the cDNA library and the cDNA fragments that were introduced into the positive clones should be provided. Or if the cDNAs of the transcription factors are cloned for screening, information about the transcription factor genes being cloned should be indicated.

Response: Thank you for suggesting providing information about Y1H screening. The information about the cDNA library was provided in Page 20, Line 423-426. In addition, the sequencing results of cDNA fragments, introduced into the positive clones, were added in Supplementary tab. 1.

Fig. 3b

I do not understand the significance of this phylogenetic tree. Is there any sense in combining different transcription factors into one phylogenetic tree?

Response: Thank you for your suggestion. After careful thought, we also considered that it did not make sense to combine different transcription factors into one phylogenetic tree and have removed this result.

Line 351 "positive"

Is this right?

Response: Sorry for any confusion caused by our description. We have revised it to a more detailed description in Page 18, Line 369-374.

The manuscript should be written more carefully because there are many minor errors.

Response: Thank you for your comments and helpful suggestions. We have revised the details of the article according to your advice.

Reviewer #2:

In this study, Jiaolei Shangguan and colleagues set out to investigate the intracellular signaling mechanisms involved in the utilization of cellulose by the fungus *Ganoderma lucidum*. First of all, cellulose is a highly abundant carbohydrate and understanding the mechanisms that take place during its degradation or conversion have direct importance to understand ecological aspects as well as a direct link to the biotechnological utilization of cellulose for biobased materials such as biofuels. On the other hand, *Ganoderma lucidum* is a mushroom-forming fungus with an important role in research and folk medicine. Altogether, the topics under study in this manuscript are very relevant.

More specifically, the authors have demonstrated a link between cellulose utilization and hydrogen sulfide (H₂S) production by fungi. This link involved the direct binding of the transcription factor CreA, known to be important for carbon catabolite repression in fungi, to the promoter of the *cbs* gene, repressing its expression. The binding site was determined too.

The production of H₂S was increased on cellulose vs. glucose, but this increase was quite modest (1.25 folds), albeit significant. Exogenously added H₂S did enhance cellulase activity, so the link seems to be solid. Furthermore, the authors conducted some validation experiments with chemicals that activate and inhibit H₂S production as well as with silenced vs. overexpression cells. All data seem to fall into place and the rationale of the work is easy to understand. Nevertheless, some of the data is somewhat circumstantial and not necessarily indicative of a direct mechanism.

Response: We are appreciated with your suggestions. Sorry for our inappropriate descriptions about partial conclusions of results. We had revised the details of the article according to your and other reviewers' advice. We hope our revised manuscript provides appropriate descriptions.

Fig.1: It would be important to present more controls for the SF7-AM method to measure intracellular accumulation of H₂S. Methods whose quantification is based on microscopy analysis strongly depend on appropriate controls so that autofluorescence is properly normalized. The same microscopy settings (e.g., exposure, gain, etc.) must be used for all samples to avoid situations of saturation, for instance. A non-staining control is missing, for instance. These controls could be a supplemental figure. Still related to this point, perhaps a more quantitative/robust method could have been tried, such as flow cytometry (in spores) or fluorimetry, using the SF7-AM probe.

Response: We are very appreciated with these important comments and suggestions. We have added the unstained control according your suggestions in the new Supplementary fig. 2.

In addition, we used the same fluorescence microscope and microscopy settings for all samples. Relative description has been added in the revised manuscript (Page 19, Line 405).

Moreover, *Ganoderma lucidum* is a multicellular filamentous fungus. There is currently very difficult to quantify it using flow cytometry methods. The current method is according to the methods used in animal cells and our previous study^{1,2}. We are also working on establishing even better methods to measure H₂S.

Reference:

¹ Tian, J. L. et al. Hydrogen sulfide, a novel small molecule signalling agent, participates in the regulation of ganoderic acids biosynthesis induced by heat stress in *Ganoderma lucidum*. *Fungal genetics and biology*, 130, 19-30, doi:10.1016/j.fgb.2019.04.014 (2019).

² Li, K. et al. Hydrogen sulfide regulates glucose uptake in skeletal muscles via S-Sulfhydration of AMPK in muscle fiber type-dependent way. *The Journal of nutrition* 153, 2878-2892, doi:10.1016/j.tjnut.2023.08.024 (2023).

Fig. 2c: In the *cbs*-silenced lines, either the gene silencing efficiency is modest or how can the authors integrate their conclusion that *cbs* is so important for cellulose degradation but at the same time *cbs*-silenced cells are able to increase CMCase activity when treated with NaHS? Perhaps justified because there are other important pathways playing a role? I think this should be discussed.

Response: Sorry for our inappropriate descriptions about the effect of H₂S on CMCase activity. We had revised the descriptions the revised manuscript (Page 18, Line 369-374).

In addition, H₂S was biosynthesized by Cystathionine β-synthase (CBS)^{1,3}. The silence of *cbs* gene resulted in a decrease in the intracellular H₂S content (Fig. 1d, e, f). NaHS is a direct donor of H₂S. The exogenous addition of NaHS can directly increase the content of H₂S without relying on the involvement of CBS. NaHS addition increased the CMCase activity in *cbs*-silenced strains. These combined results suggest that H₂S improved the cellulase activity of *G. lucidum* under cellulose culture condition.

We also added to the discussion other underlying regulators participate in the improvement of CBS to cellulose utilization in *G. lucidum* in Page 15-16, Line 304-320. We hope that this is conducive to understanding the effect of H₂S in regulation of cellulose utilization.

Reference:

¹ Tian, J. L. et al. Hydrogen sulfide, a novel small molecule signalling agent, participates in the regulation of ganoderic acids biosynthesis induced by heat stress in *Ganoderma lucidum*. *Fungal genetics and biology*, 130, 19-30, doi:10.1016/j.fgb.2019.04.014 (2019).

³ Landry, A. P., Roman, J. & Banerjee, R. Structural perspectives on H₂S homeostasis. *Current opinion in structural biology*, 71, 27-35, doi:10.1016/j.sbi.2021.05.010 (2021).

Fig. 2d: differences are quite modest, despite the statistical significance.

Response: We appreciate with your suggestions. We had revised the unclear descriptions in the revised manuscript (Page 9, Line 173). We hope our revised manuscript provides appropriate descriptions about the effect of H₂S in regulation of cellulose utilization.

Fig. 3b: the phylogenetic is not showing informative data. Many more protein accessions would have to be used to make a more detailed study of the relationship between these transcription factors. Furthermore, the description of this figure in the legend is “evolutionary tree”, which is an incorrect oversimplification. Additionally, the values next to the tree branches are not defined in the legend.

Response: Thank you very much for your advice. According your suggestion, we have deleted the phylogenetic tree.

Minor changes

- Line 41: “maximum amount of renewable carbohydrates”; this phrase is unclear. What is meant by “maximum”?

Response: Thank you for your questions about our inaccurate description. We have revised according your suggestions (Page 3, Line 39-40).

- Line 76: “H₂S improved the tolerance of plants to Cu²⁺ and heat”; not clear or not specific enough. How did H₂S improve tolerance?

Response: According to the suggestions, we have added a specific description of mechanism of H₂S improve the tolerance of plants to Cu²⁺ and heat in Page 5, Line 84-85.

- Line 82: the word “well” seems unnecessary.

Response: We have removed “well” according your advice in Page 5, Line 91.

- Line 95: “a gene of H₂S synthesis enzyme”; this phrase is not properly written. Perhaps “a gene encoding the H₂S synthesis enzyme”? But could the authors be more specific about the role of this enzyme in the synthesis of H₂S?

Response: According your advice, we have revised the description in Page 5, Line101-102 and added the role of CBS in H₂S synthesis in Page 4, Line 76-77.

- Line 97: “which promoting” is incorrect. “which promotes”?

Response: We have revised accordingly in Page 6, Line 104.

- Fig. 2E: the meaning of the red, horizontal line should be specified in the figure legend.

Response: We have illustrated the meaning of the red horizontal line in the Figure legends of Fig.2 and Fig.6 (Page 24, Line 495-496 and Page 25, Line 532-533).

- Line 287-305 seems more appropriate for the Introduction rather than Discussion

Response: We have moved this part into the introduction (Page 3-4, Line 59-65) according your suggestion.

The manuscript is well written and easy to understand. However, an overall revision could be performed to fix minor incorrections.

Response: Thank you very much for your suggestions. Those comments are valuable and very helpful for our paper, and we have revised the details of the article according to your advice.

Pedro Gonçalves

Reviewer #3:

General Comments:

The manuscript presents a well-executed study investigating the role of the hydrogen sulfide (H₂S) signaling pathway in *Ganoderma lucidum*, particularly its impact on cellulase activity and cellulose utilization. The research is meticulously conducted, and the findings contribute valuable insights into the molecular mechanisms underlying microbial cellulose utilization. The presented data is well-supported and convincingly demonstrates the involvement of the H₂S signaling pathway, specifically regulated by the carbon repressor transcription factor CreA.

Specific Comments:

Clarity and Structure:

The manuscript is well-structured, and the clarity of the presentation is commendable. The introduction effectively contextualizes the study, and the objectives are clearly defined. The results and discussion sections are logically organized, facilitating a smooth understanding of the experimental outcomes.

Experimental Design and Methods:

The experimental design is robust, and the methods are meticulously detailed, allowing for reproducibility. The use of both pharmacological and genetic experiments to probe the H₂S signaling pathway, as well as the involvement of CreA, enhances the study's credibility.

Data Presentation:

The data presentation is clear and concise. Figures and tables are appropriately utilized to illustrate key findings, and the inclusion of statistical analyses adds rigor to the interpretation of results. The use of fluorescence probes and gene expression analysis provides a comprehensive view of H₂S concentration and its regulatory mechanisms.

Discussion of Results:

The discussion of results is comprehensive, and the authors effectively relate their findings to existing literature. The identification of CreA as a negative regulator of H₂S biosynthesis adds novelty to the study. Additionally, the discussion on the potential influence of CBS on cellulose utilization is insightful.

Language and Style:

The language used in the manuscript is generally clear and concise. However, attention to minor grammatical issues, as identified in the detailed suggestions, will enhance the overall polish of the manuscript.

Conclusion:

The study significantly contributes to our understanding of the molecular mechanisms governing cellulose utilization in *G. lucidum*. The data is robust, the methodology is sound, and the results are well-discussed. With minor revisions for language and clarity, the manuscript is well-suited for publication in a peer-reviewed journal.

Response: We appreciate the positive comments about the manuscript.

Here are some minor issues that needs to be addressed.

Some references regarding CreA homologs in other organisms should be included for example PMID: 21980519, PMID: 21619626, PMID: 31040248.

Response: Thank you very much for your advice. According your suggestion, we have added the references in Page 4, Line 62-64 and Page 17, Line 344-348.

Here are some minor points with language use.

In line 104, "*Ganoderma lucidum*" could be italicized or written in a different format for emphasis on the species name.

Response: We have italicized the "*Ganoderma lucidum*" in Page 6, Line 111 according your suggestion. The relevant issues have been examined in the whole revised manuscript.

In line 108, it should be "This result shows that" instead of "This result shown that."

Response: Thank you very much for your suggestion. We have revised tense according to your advice (Page 6, Line 115).

In line 111, "genes transcription levels" should be "gene transcription levels."

Response: We have revised according to your suggestion (Page 6, Line 118).

In line 116, "cystathionine β -synthase (CBS)" could be introduced with "the" for better clarity, like "the cystathionine β -synthase (CBS)."

Response: We have added "the" in line (Page 6, Line 123). The relevant issues have been examined in the whole revised manuscript.

In line 118, "*cbs*-silenced and *cbs*-overexpressed strains" could be rewritten as "*cbs*-silenced, *cbs*-overexpressed, and control strains" for better clarity.

Response: We have revised the relevant issues in the whole revised manuscript (Page 7, Line 129; Page 8, Line 157-158 and Page 11, Line 223).

In line 123, "H₂S biosynthesis in *G. lucidum* and CBS might be one" might read more smoothly as "H₂S biosynthesis in *G. lucidum*, and CBS might be one."

Response: Thank you very much for your advice. We have revised according your suggestion in Page 7, Line 140.

In line 126, "To explore" could be followed by a phrase indicating the action, such as "To explore the effect of H₂S signal on cellulose utilization."

Response: We have revised according your suggestion in the revised manuscript (Page 7, Line 143-144).

In line 136, "These results of pharmacological experiments suggested that" could be more concise, like "Pharmacological experiments suggested that."

Response: We have revised according your advice in Page 8, Line 154.

In line 158, there's a space before the colon in "regulators of the cbs gene identifies diverse transcription." It should be "regulators of the cbs gene identifies diverse transcription:"

Response: The space before the colon in "regulators of the cbs gene identifies diverse transcription." has been deleted.

In line 218, "which indicated that cellulose culture condition" could be revised for clarity, like "indicating that cellulose culture conditions."

Response: We are appreciated with your suggestions. We have revised according your advice in the revised manuscript (Page 12, Line 237).

In line 234, "thereby improving cbs transcription and H₂S biosynthesis under cellulose culture condition" could be simplified, such as "thereby enhancing cbs transcription and H₂S biosynthesis under cellulose culture conditions."

Response: We have simplified the description according your advice in Page 12, Line 253-254.

In line 270, "Microbial cellulose utilization contributes to the carbon cycle in the biosphere and is widely used in both agriculture and industry" could be split into two sentences for clarity.

Response: Thank you for your suggestion. We have splited into two sentences to descript according your advice (Page 14, Line 288-289).

REVIEWERS' COMMENTS:

Reviewer #1 (Remarks to the Author):

The authors have carefully responded to my comments. As the authors claim, this manuscript clearly shows that H₂S synthesis mediated by CBS gene expression is regulated by CreA, a well-known transcription repressor in filamentous fungi. I understand the difficulty in finding out how H₂S affects cellulase activity, but I think more information is needed on the relationship between CreA and H₂S affecting cellulase activity. For instance, it is interesting that double knockdown of creA and cbs reduces CMCase activity relative to WT, even though creA knockdown would be expected to increase cellulase gene expression (Fig. 6a). The relationship between CreA and H₂S in cellulase activity would be more clear if the expression of cellulase genes were examined.

Fig. 1b

I don't understand the y-axis label; what does 100% indicate?

Response: Thank you for your questions. We defined the fluorescence intensity value of Sulfidefluor-7 acetoxymethyl ester (SF7-AM) detected under glucose culture condition as 100%. Relative quantification was used to calculate the fluorescence values of SF7-AM detected under cellulose culture conditions. Corresponding description has been added in the Figure legends (Page 23, Line 474-475).

I still don't understand what 100% of the y-axis means. I understand that the fluorescence intensity value of SF7-AM detected under glucose culture conditions is 100%. However, the values on the y-axis are relative values, not % values. I think the (100%) on the y-axis label is misleading to the reader.

Additional points

Is the number of significant digits appropriate? I think the percentage and fold values should be rounded to the first decimal place.

What do the blue borders and the light red rounded squares in Fig. 7 represent?

Reviewer #2 (Remarks to the Author):

I have now reviewed the revised version of the manuscript submitted by Shangguan et al.

The authors have appropriately addressed my analyses. I maintain my previous opinion that the data does not fully support a direct mechanism. Nevertheless, this is, of course, not always possible to achieve, and it does not reduce the relevance of the study.

As the authors properly suggest in their response, improved methodologies to investigate the role of H₂S in fungi will benefit future studies and conclusions.

In conclusion, there is substantial merit and interest in this research work.

Reviewer #3 (Remarks to the Author):

The authors have improved the MS significantly and the most of the concerns of this reviewer has been addressed.

Dear Reviewers:

We appreciate the positive comments about our manuscript entitled “**CBS/H₂S signalling pathway regulated by the carbon repressor CreA promotes cellulose utilization in *Ganoderma lucidum***” (COMMSBIO-23-3923). Those comments are valuable and very helpful. We have revised our manuscript according to your and reviewers' suggestions.

The revised sections are highlighted in red color in the revised manuscript.

Our point-by-point replies to the reviewers' comments are presented below.

Thank you very much for your attention and consideration. We hope that the revised manuscript will be appropriate for publication.

Best regards.

Yours sincerely,

Mingwen Zhao

E-mail: mwzhao@njau.edu.cn

< COMMENTS FOR THE AUTHOR >

Reviewer #1 (Remarks to the Author):

The authors have carefully responded to my comments. As the authors claim, this manuscript clearly shows that H₂S synthesis mediated by *CBS* gene expression is regulated by CreA, a well-known transcription repressor in filamentous fungi. I understand the difficulty in finding out how H₂S affects cellulase activity, but I think more information is needed on the relationship between CreA and H₂S affecting cellulase activity. For instance, it is interesting that double knockdown of *creA* and *cbs* reduces CMCase activity relative to WT, even though *creA* knockdown would be expected to increase cellulase gene expression (Fig. 6a). The relationship between CreA and H₂S in cellulase activity would be more clear if the expression of cellulase genes were examined.

Response: We are very appreciated with your professional comments and suggestions. We also think the phenomenon that the CMCase activity (Fig. 6a) in *creA-cbs-silenced* strains were significantly reduced compared with that in *wt* strains, while the CMCase activity (Fig. 6a) and the cellulase gene expression¹ in *creA*-silenced strains was significantly increased compared with that in the *wt* strain is very interesting. These results imply an important regulatory role of CBS-synthesized H₂S in cellulase activity, and there are various potential mechanisms involved. However, due to the tight schedule, we only have added the discussion about underlying mechanisms by which H₂S affecting cellulase activity in the revised manuscript (Page 15-16, Line 319-345) according to your and editor's advice. We are also working on explore the underlying mechanisms by which H₂S affecting cellulase activity *G. lucidum*.

¹ Hu, Y. et al. In *Ganoderma lucidum*, Glsnf1 regulates cellulose degradation by inhibiting GICreA during the utilization of cellulose. Environmental microbiology 22, 107-121, doi:10.1111/1462-2920.14826 (2020).

I don't understand the y-axis label; what does 100% indicate?

Response: Thank you for your questions. We defined the fluorescence intensity value of Sulfidefluor-7 acetoxymethyl ester (SF7-AM) detected under glucose culture condition as 100%. Relative quantification was used to calculate the fluorescence values of SF7-AM detected under cellulose culture conditions. Corresponding description has been added in the Figure legends.

I still don't understand what 100% of the y-axis means. I understand that the fluorescence intensity value of SF7-AM detected under glucose culture conditions is 100%. However, the values on the y-axis are relative values, not % values. I think the (100%) on the y-axis label is misleading to the reader.

Response: We are very appreciated with the suggestion. According to your advice, we have deleted the (100%) on the y-axis label in Fig. 1b, 1e, 4e and 5d. The corresponding descriptions have been deleted in the Figure legends.

Additional points

Is the number of significant digits appropriate? I think the percentage and fold values should be rounded to the first decimal place.

Response: Thank you for your suggestion. We have revised the number of significant digits according to your advice in the revised manuscript (Page 2, Line 28-31 and Page 6-13, Line 114-282).

What do the blue borders and the light red rounded squares in Fig. 7 represent?

Response: We are appreciated with your question. We have added the corresponding description in Fig. 7.

Reviewer #2 (Remarks to the Author):

I have now reviewed the revised version of the manuscript submitted by Shangguan et al.

The authors have appropriately addressed my analyses. I maintain my previous opinion that the data does not fully support a direct mechanism. Nevertheless, this is, of course, not always possible to achieve, and it does not reduce the relevance of the study.

As the authors properly suggest in their response, improved methodologies to investigate the role of H₂S in fungi will benefit future studies and conclusions.

In conclusion, there is substantial merit and interest in this research work.

Response: We appreciate the positive comments about the manuscript.

Reviewer #3 (Remarks to the Author):

The authors have improved the MS significantly and the most of the concerns of this reviewer has been addressed.

Response: We appreciate the positive comments about the manuscript.

** See the Nature Portfolio author and referees' website at www.nature.com/authors for information about policies, services and author benefits

This email has been sent through the Springer Nature Tracking System NY-610A-NPG&MTS

Confidentiality Statement:

This e-mail is confidential and subject to copyright. Any unauthorised use or disclosure of its contents is prohibited. If you have received this email in error please notify our Manuscript Tracking System Helpdesk team at <http://platformsupport.nature.com> .

Details of the confidentiality and pre-publicity policy may be found here <http://www.nature.com/authors/policies/confidentiality.html>

Privacy Policy | Update Profile

DISCLAIMER: This e-mail is confidential and should not be used by anyone who is not the original intended recipient. If you have received this e-mail in error please inform the sender and delete it from your mailbox or any other storage mechanism. Springer Nature America, Inc. does not accept liability for any statements made which are clearly the sender's own and not expressly made on behalf of Springer Nature America, Inc. or one of their agents.

Please note that neither Springer Nature America, Inc. or any of its agents accept any responsibility for viruses that may be contained in this e-mail or its attachments and it is your responsibility to scan the e-mail and attachments (if any).